# Culm Morphological Analysis in Moso Bamboo Reveals the Negative Regulation of Internode Diameter and Thickness by Monthly Precipitation

**DOI:** 10.3390/plants13111484

**Published:** 2024-05-28

**Authors:** Qianwen Zhang, Xue Chu, Zhipeng Gao, Yulong Ding, Feng Que, Zishan Ahmad, Fen Yu, Muthusamy Ramakrishnan, Qiang Wei

**Affiliations:** 1State Key Laboratory of Tree Genetics and Breeding, Co-Innovation Center for Sustainable Forestry in Southern China, Bamboo Research Institute, Key Laboratory of National Forestry and Grassland Administration on Subtropical Forest Biodiversity Conservation, School of Life Sciences, Nanjing Forestry University, Nanjing 210037, China; zhangqianwen2658@163.com (Q.Z.); ylding@vip.163.com (Y.D.);; 2Jiangxi Provincial Key Laboratory for Bamboo Germplasm Resources and Utilization, Jiangxi Agriculture University, Nanchang 330045, China; yufen@jxau.edu.cn

**Keywords:** Moso bamboo, culm, internode, phenomics, phenome, populations, morphological analysis

## Abstract

The neglect of Moso bamboo’s phenotype variations hinders its broader utilization, despite its high economic value globally. Thus, this study investigated the morphological variations of 16 Moso bamboo populations. The analysis revealed the culm heights ranging from 9.67 m to 17.5 m, with average heights under the first branch ranging from 4.91 m to 7.67 m. The total internode numbers under the first branch varied from 17 to 36, with internode lengths spanning 2.9 cm to 46.4 cm, diameters ranging from 5.10 cm to 17.2 cm, and wall thicknesses from 3.20 mm to 33.3 mm, indicating distinct attributes among the populations. Furthermore, strong positive correlations were observed between the internode diameter, thickness, length, and volume. The coefficient of variation of height under the first branch showed strong positive correlations with several parameters, indicating variability in their contribution to the total culm height. A regression analysis revealed patterns of covariation among the culm parameters, highlighting their influence on the culm height and structural characteristics. Both the diameter and thickness significantly contribute to the internode volume and culm height, and the culm parameters tend to either increase or decrease together, influencing the culm height. Moreover, this study also identified a significant negative correlation between monthly precipitation and the internode diameter and thickness, especially during December and January, impacting the primary thickening growth and, consequently, the internode size.

## 1. Introduction

Bamboo, a sustainable non-timber forest product with around 10,000 documented applications [1,2,3], also contributes to mitigating climate change by sequestering large quantities of carbon [4,5,6,7]. Among various bamboo species, Moso bamboo (*Phyllostachys edulis* (Carrière) J. Houz) stands out as the fastest-growing, tree-like plant species, achieving a height of over 20 m in vertical stem growth in just 45–60 days during its rapid growth phase, with a peak growth rate of more than 1 m per day [8]. Widely distributed across East and Southeast Asia, Moso bamboo covers 4.43 million hectares in China and serves as a key raw material for the bamboo industry, holding significant economic and ecological importance [9,10]. Nevertheless, in recent years, the total area of Moso bamboo forest cultivation has steadily increased to over 5.27 million hectares, driven primarily by its growing value in international trade [11,12].

However, biotic and abiotic stress significantly hamper Moso bamboo cultivation [13]. Consequently, considerable progress has been achieved in understanding its molecular stress responses [14,15,16,17,18,19]. Through this research, numerous stress-related genes, including circular RNAs, have been identified in Moso bamboo under stress conditions [20,21,22,23,24,25,26,27]. These findings suggest that Moso bamboo exhibits an intricate adaptation mechanism to modify phenotypic traits when faced with stress conditions. Additionally, understanding its fast growth process is crucial for rapid biomass production [28,29,30]. Thus, significant improvement has been made in understanding its culm development, which is a well-defined process involving various stages [8,31,32,33,34,35,36,37]. Therefore, understanding the transitions between these stages is crucial to comprehend the mechanisms driving its rapid growth. The studies suggest that the rapid growth mechanism is controlled at multiple levels, with hormones like auxin (IAA) and gibberellin (GA) playing crucial roles, along with their downstream target genes related to cell wall biogenesis [8,33,38,39,40,41,42]. Additionally, microRNAs (miRNAs) and DNA and RNA methylations regulate the expression of genes associated with DNA replication, cell division, hormonal responses, poly(A) tail lengths, 3′ UTR (untranslated region), and alternative splicing, contributing to the molecular mechanisms governing rapid growth [41,43].

Recently, in our in-depth study [8], we also identified internode 18 as a representative internode for rapid growth, with culms capable of growing at a rate of 114.5 cm/day. This internode includes a 2 cm cell division zone, a cell elongation zone up to 12 cm, and a secondary cell wall thickening zone (lignification). The study suggests that gibberellin may directly trigger the rapid growth, while both decreased cytokinin and increased auxin accumulation may trigger the elongation of the cell division zone [8]. Additionally, abscisic acid and mechanical pressure may rapidly stimulate secondary cell wall thickening. The study also developed a growth model for internode 18 that includes 14 developmental stages representing critical transition points in cell division, cell elongation, and secondary cell wall thickening [8]. The growth rate of internode 18 also greatly exceeds the vegetative growth reported in other plants.

Despite advancements in understanding the mechanism regulating rapid growth, in our previous study [8], we identified internode 18 as the longest among the internodes in Moso bamboo populations. This determination was made through the analysis of more than 12,750 internodes from over 510 culms in 17 Moso bamboo populations grown in the major centers of bamboo diversity in China, with each population comprising 30 mature bamboo culms. Additionally, the studies suggested that internode length and functional traits may be influenced by the environmental temperature [8,44]. The studies further suggest that a stable new dwarf variant of Moso bamboo (*P. edulis* f. *exaurita* T. G. Chen) exhibits a considerable reduction in height compared to the wild-type Moso bamboo, with the variant measuring only 49% of the height of the wild-type. This dwarf variant shows a decrease in both the number and length of internodes compared to the wild-type, contributing to its dwarfism. This phenomenon is attributed to a reduction in both cell number and cell length within the internode [45]. In addition, the reduced culm diameter, shortened internode length, and diminished total number of internodes significantly decreased fresh weight, contributing to the reduced biomass and morphological development of the bamboo culm [46]. The mechanical properties of Moso bamboo culms are also significantly influenced by various environmental factors, including Moso bamboo populations, growth characteristics, and the growth environment [47].

Consequently, understanding population genome information and culm morphological variations across different Moso bamboo populations is crucial for unraveling the intricacies of internode developments and bamboo cultivation and utilization. Moreover, they contribute significantly to understanding the health status of Moso bamboo forests. Such understanding is paramount for ensuring ecological, socio-economic, and environmental sustainability [48]. However, little attention has been paid to this aspect. A recent study [49] uncovered a genomic variation atlas of Moso bamboo by identifying 5.45 million single-nucleotide polymorphisms (SNPs) through whole-genome resequencing of 427 individuals across 15 representative geographic areas. The study also revealed that low genetic diversity, high genotype heterozygosity, and genes under balancing selection drive Moso bamboo population adaptation. Moreover, another recent study [50] integrated the SNPs of 20 Moso bamboo forms, primarily characterized by distinct culm shape variations, with the SNPs of these 427 individuals and identified low genetic diversity and high genotype heterozygosity associated with the formation of Moso bamboo forms. However, despite these advancements, there remains a gap in our understanding of culm morphological variations across different Moso bamboo populations, which is crucial for selecting suitable bamboo varieties for specific purposes. Therefore, the present study aimed to comprehensively explore culm morphological variations, including internode length, internode diameter, height under the first branch, number of internodes, culm height, and wall thickness, within various Moso bamboo populations in China.

## 2. Results

### 2.1. Culm Heights

To analyze culm morphological variations among different Moso bamboo populations in China, various parameters of the culms and internodes were examined. The samples were collected from 16 major bamboo-producing areas in China. Each of these sampling regions represents a distinct Moso bamboo population identified by an abbreviation code, such as AJ, CS, CY, DA, HS, JZ, LY, RH, TJ, WYS, XA, XN, YA, YF, YX, and ZT, as established in our previous study [8]. A geographical map illustrating the distribution of populations and sampling regions is provided in Appendix A. The culm height and internode length data utilized in this study were previously published in our previous study [3]. Additionally, we investigated different parameters in the current study. Thirty healthy, uniform culms were randomly selected from each sampling region, with a distance of 100 m between each culm, based on the parameters of the standard culm diameter at breast height (DBH).

The culms had a height of more than 18 m, and the basal internode was relatively short. The upper internode gradually increased, with the middle internode measuring more than 40 cm in length, representing the main growth phase. The average culm heights across the various populations ranged from 9.67 m to 17.5 m, with the population AJ exhibiting the lowest height and the population CS having the highest length (Appendix A). The coefficient of variation provided insights into the consistency of culm heights within each population, with higher values suggesting greater variability.

### 2.2. Culm Heights under the First Branch

The average culm heights under the first branch ranged from 4.91 m to 7.67 m across the various populations, with the population AJ exhibiting the lowest height, and the population YA having the highest length (Appendix A). The range and coefficient of variation suggest that, compared to total culm heights, the culm heights under the first branch exhibited higher variability within each population. The population AJ had the lowest mean values for both total culm height and culm heights under the first branch. Conversely, the population CS had the highest mean value in total culm height, with a mean value of 17.5 m (Appendix A), while the population YA had a slightly lower mean value of 16.1 m (Appendix A). Similarly, in terms of culm heights under the first branch, the population CS again had the highest mean value at 7.33 m, followed closely by the population YA with a mean value of 7.67 m. This consistent pattern demonstrates that the population CS consistently had higher values for both total culm height and culm heights under the first branch.

### 2.3. Ratios of Culm Heights under the First Branch to the Total Culm Height

To understand the relationship and contribution between the culm heights under the first branch and the total culm height, their ratio was calculated. The minimum ratio was observed in the population YA, while the maximum was found in the population TJ (Appendix A). The ratio across populations ranged from 0.39% to 0.53%. The population RH exhibited the highest mean ratio (0.53%), indicating a significant contribution of the first branch to the total height. In contrast, the populations CY, JZ, and LY displayed the lowest mean ratio (0.39%), indicating a relatively reduced contribution to the total height. The coefficient of variation indicated the relative contribution of culm heights under the first branch to the total culm height across different populations, with higher coefficient of variation values suggesting notable variability in the ratio within each population.

### 2.4. Internode Number under the First Branch

The number of internodes ranged from 17 to 36, with the average internodes ranging from 20.6 to 30.9. The population YA had the highest average (30.9), while the population AJ had the lowest average (20.6) (Appendix A). Notably, these average internode numbers aligned with the average culm heights under the first branch in the populations AJ (lowest length) and YA (highest length). In contrast, although the population CS showed higher values for both total culm height and culm heights under the first branch, it did not have the lowest or highest average total number of internodes (25.2). Within each population, the range of the total number of internodes varied from 8.0 to 12, with the coefficient of variation ranging from 8.51% to 12.5%. This suggests that each population demonstrates moderate variability in the total number of internodes under the first branch, indicating genetic diversity and distinct features within each population.

### 2.5. Internode Lengths

To measure the length of internodes across all the populations, a total of 11,548 internodes were recorded, with each population ranging from 619 to 928 internodes and an average of 721.75. The minimum and maximum internode lengths ranged from 0.80 cm to 46.4 cm. Notably, the population XN had the highest maximum internode length (46.4 cm), while the population TJ had the shortest minimum length (0.80 cm) (Appendix A). The range of internode lengths within each population demonstrates significant variation (29.3 cm to 42.5 cm), with population TJ exhibiting the widest range of 42.5 cm. In addition to displaying higher values for both total culm height and culm heights under the first branch, the population CS also exhibited the highest average internode length (27.8 cm). In contrast, the population CY had the shortest average internode length (21.4 cm). The coefficient of variation emphasized moderate variability in internode lengths within each population, underscoring the genetic diversity and distinct characteristics present.

### 2.6. Internode Diameters

The internode diameter values ranged from 5.10 cm to 17.2 cm across the populations, with the minimum internode diameter recorded in the population YA and the maximum in the population CS (Appendix A). The population CS exhibited the highest average internode diameter (11.1 cm), followed by the populations WYS (10.5 cm) and JZ (10.4 cm), indicating superior growth. Conversely, the population ZT had the lowest average diameter (8.53 cm). The standard deviation ranged from 0.97 cm to 1.68 cm, indicating the degree of variation in internode diameters within each population. The coefficient of variation indicated moderate to high variability in internode diameters within most populations. The populations YA, DA, LY, RH, WYS, HS, and XN showed relatively higher variability, while the population ZT exhibited lower variability.

### 2.7. Internode Wall Thicknesses

The internode wall thickness varied considerably among the populations, ranging from the thickest to the thinnest walls. The populations JZ and YX exhibited a minimum wall thickness of 3.20 mm, while the population RH recorded a maximum wall thickness of 33.3 mm (Appendix A). The range of internode wall thickness within each population varied from 10.3 mm to 27.7 mm, with the population RH showing the widest range, indicating that greater thickness resulted in wider ranges within the population. On average, the population CS had the highest internode wall thickness of 11.2 mm, followed closely by the population WYS with 11.0 mm. In contrast, the population AJ exhibited the lowest average wall thickness of 9.11 mm. Among the populations, CS and WYS generally displayed thicker internode walls, measuring greater than 11 mm, while DA, HS, CY, RH, JZ, and LY showed moderate wall thicknesses ranging from 10.0 mm to 10.4 mm, and others had thinner walls ranging from 9.11 mm to 9.98 mm, highlighting variability in wall thickness across the populations.

### 2.8. Correlation Analysis between Culm Parameters

To predict the relationships between culm parameters and determine how they influence culm heights, the correlation analysis was conducted using the mean values of the parameters (Appendix A). Both positive and negative correlations indicate how the parameters tend to increase or decrease together. The internode diameter (D) was positively correlated with the internode thickness (T), with a correlation coefficient of 0.69. In contrast, the diameter had a strong negative correlation with the diameter slope (D Slope) with a correlation coefficient of −0.85 (Table 1). The thickness slope (T Slope), showed no significant correlations with the other parameters. Furthermore, the diameter and thickness slope (DT Slope) exhibited negative correlations with the diameter, thickness, and diameter slope. The diameter and length slope (DL Slope) and the thickness and length slope (TL Slope) showed moderate correlations with the other parameters. The internode length (L) was positively correlated with only four parameters: culm height (H), height under the first branch (HUFB), internode volume (IV), and the coefficient of variation of height under the first branch (CVHUFB). The internode volume refers to the space occupied by a bamboo internode, typically cylindrical in shape and defined by its enclosed volume.

The culm height demonstrated positive correlations with the diameter, thickness, height under the first branch, internode volume, and CVHUFB, with correlation coefficients ranging from 0.66 to 0.89, suggesting the importance of culm heights under the first branch. Additionally, the height under the first branch exhibited positive correlations with the diameter, culm height, internode number under the first branch (INUFB), internode volume, and CVHUFB. In contrast, the internode number under the first branch showed weak to moderate correlations with the other parameters. The internode volume had a strong positive correlation with the diameter, thickness, internode length, culm height, and height under the first branch (Table 1).

Likewise, the CVHUFB demonstrated strong positive correlations with the diameter, thickness, internode length, culm height, height under the first branch, internode number under the first branch, and internode volume. However, both the ratio of first branch to culm height (Ratio FBRS) and maximum volume internode number (Vmax IS) showed no positive correlation with the other parameters. Furthermore, both the maximum volume internode number (Vmax in ratio) and maximum volume height between nodes (Vmax H ratio) showed strong positive correlations with the diameter and thickness slope, with correlation coefficients of 0.81 and 0.80, respectively, and had a negative correlation with the culm height, with correlation coefficients of −0.52 and −0.53, respectively.

Based on the correlation analysis, both the diameter and thickness are significant parameters contributing to the internode volume, culm height, and CVHUFB, highlighting their crucial roles in determining the overall height of Moso bamboo culms. However, the diameter appears to have a slightly stronger association with culm height compared to thickness, based on the correlation coefficients (Table 1). Therefore, diameter may be considered slightly more important in determining culm height in Moso bamboo.

### 2.9. Culm Parameters Tend to Increase or Decrease Together to Influence Culm Height

To understand how the covariation of culm parameters, where they tend to increase or decrease together, influences culm height, the coefficient of determination (*r*^2^) was calculated using the culm parameters. Both the internode length and internode order showed a tendency to increase together across all the populations, with *r*^2^ values nearly reaching 1.0 (ranging from 0.968 to 0.999) (Figure 1), indicating a strong positive correlation where an increase in one variable corresponds to an increase in the other. In contrast, the internode diameter exhibited a tendency to decrease while the internode order displayed a tendency to increase across all the populations, with *r*^2^ values ranging from 0.988 to 0.999 and a slope value ranging from −0.131 to −0.192, indicating a negative slope, without particularly steep declines, suggesting a strong negative correlation between the variables. The slope value indicates the average change in the dependent variable (internode diameter) for a one-unit increase in the independent variable (internode order). For instance, with a slope value of −0.192, for every one-unit increase in the internode order, the internode diameter decreased by 0.192 units on average (Appendix A). Similarly, the internode thickness exhibited a tendency to decrease while the internode order exhibited a tendency to increase across all the populations (Figure 2). Additionally, as the internode thickness tended to increase, the internode length exhibited a tendency to decrease across all the populations (Figure 3).

The relationship between the internode volume and internode order suggests how the internode volumes change along the culm height. As the internode order increases, indicating the position along the culm height, a distinct pattern is observed in the internode volume (Figure 4). Initially, it tends to decrease at the bottom of the culm, then increases towards the middle, and finally decreases again towards the top of the culm. This pattern indicates variations in the internode size along the culm height. The mean internode order provides insight into where, on average, the highest internode volume occurs along the culm height, with the mean internode order values ranging from 9.5 to 14.9 across all the populations.

The relationship among the internode length, thickness, and diameter exhibited strong correlation coefficients. The hillslope value, ranging from 0.16 to −0.85, indicates the rate of change between variables. A positive hillslope value suggests an increasing trend (positive slope), while a negative value suggests a decreasing trend (negative slope). For example, as observed in the AJ population, when the internode diameter increases, the internode length decreases, with a steep negative hillslope value of −0.85. Conversely, when the internode diameter increases, the internode thickness increases, with a less steep positive hillslope value of 0.44 in the same AJ population (Figure 5). Across all the populations, the positive slope values ranging from 0.16 to 0.44 represent the internode thickness increase, while the negative slope values ranging from −0.27 to −0.85 represent the internode length decrease. According to these hillslope values, the internode length decreases and the internode thickness increases, following relatively smooth and gradual slopes, rather than steeper ones.

Altogether, as the internode order increases, the internode diameter decreases, with an *r*^2^ value of 0.997 and a negative slope value of −0.1665, indicating a gradual decline rather than a steep decline (Figure 6a). The coefficient of variation of the internode order falls between >0.05 and <0.10 (Figure 6b). Similarly, as the internode order increases, the internode thickness decreases (Figure 6c), with the coefficient of variation of the internode order falling between >0.05 and <0.15 (Figure 6d). Notably, when the internode diameter increases, the internode length decreases (Figure 6e). Conversely, when the internode diameter increases, the internode thickness increases (Figure 6e). Likewise, as the internode thickness increases, the internode length decreases (Figure 6f). These results indicate that Moso bamboo follows a pattern of gradual decline and increase rather than steep declines and increases, analogous to a building construction where the foundation is typically designed to be robust to support the structure, with the load gradually decreasing as the number of floors increases. Similarly, Moso bamboo exhibits a comparable pattern, where the base of the culm is strong to support the plant’s weight, with a gradual decrease in internode size and volume as it grows taller.

### 2.10. Influence of Environmental Factors on Internode Diameter and Thickness

The correlation analysis revealed that both internode diameter and thickness are significant parameters contributing to the culm height. Therefore, to understand the influence of environmental factors on the internode diameter and thickness, the Pearson correlation between environmental factors and the internode diameter and thickness was calculated. Subsequently, the internode diameter and thickness were compared with monthly temperature (Figure 7a), precipitation (Figure 7b), and humidity (Figure 7c). Interestingly, while the temperature exhibited some correlation with both the internode diameter and thickness, these correlations did not reach statistical significance (Figure 7a). Similarly, the humidity displayed varying degrees of correlation with both the internode diameter and thickness, yet none were significant (Figure 7c). Remarkably, among these factors, only precipitation emerged with a significant negative correlation with both the internode diameter and thickness. Notably, this significant negative correlation was observed predominantly in the months of December and January (Figure 7b), suggesting the impact of precipitation on the primary thickening growth.

Before culm development, Moso bamboo typically undergoes various developmental stages, including the primary thickening growth of the underground shoot bud from the rhizome [51]. This primary thickening growth phase usually occurs from December to January. At this stage, the bud becomes an active and mature shoot with distinguishable internodes and nodes, and the pith cavity formation also occurs simultaneously, which plays an important role in determining the internode sizes [31,32]. As the primary thickening growth occurs underground, precipitation impacts the primary thickening growth, thereby affecting both the internode diameter and thickness.

## 3. Discussion

### 3.1. Bamboo Culm Material

Moso bamboo is widely distributed and extensively utilized in the bamboo industry [52], but its adaptability varies across different regions of China [47,53]. Therefore, investigating culm morphological variations among different Moso bamboo populations provides valuable genetic insights and identifies adaptations to diverse environmental conditions [46]. Despite being capable of sexual reproduction, Moso bamboo predominantly spreads through asexual reproduction [51]. Nevertheless, its irregular flowering further restricts sexual reproduction, limiting population development and impeding the understanding of culm ages. Consequently, identifying germplasm resources and ensuring the availability of uniformly healthy culms across all populations present significant challenges. As a result, the culms developed through asexual reproduction are widely utilized and were considered populations for studying culm morphological variations [8,47]. A recent study [50], integrating the SNPs of 20 Moso bamboo forms with those of 427 wild-type Moso bamboo individuals from 15 geographic regions, each representing a distinct population, uncovered low genetic diversity and high genotype heterozygosity associated with the formation of Moso bamboo forms. Asexually propagated Moso bamboo can tolerate more somatic mutations compared to plants propagated purely through sexual reproduction [50]. Consequently, both wild-type Moso bamboo and its forms are characterized by low-frequency and high-frequency heterozygous genotypes. Additionally, vast morphological differences exist between the wild-type and forms of Moso bamboo. These variations encompass culm shape, culm color, and other characteristic differences, each defining distinct categories of variation [50]. Nonetheless, the culm morphological variations among the populations remain unknown or receive limited attention, further restricting broader bamboo culm utilization. Therefore, this study investigated the culm morphological variations of over 480 culms collected from sixteen Moso bamboo populations in China.

Furthermore, the height, volume, and thickness of Moso bamboo culms remain relatively stable after the completion of the rapid growth, but the moisture content decreases significantly. The formation and growth of secondary walls in the culms are mainly reflected in the cell-wall thickness, which gradually increases with the age of the culms. The dry matter in the culm wall increases year by year with the aging of the culms, reaching its peak between 6 and 8 years of age, after which it begins to decrease. Additionally, after 3 to 5 years of rapid growth, the culm material reaches its optimal state with the highest strength. However, upon reaching a certain age (12 years), the culms show signs of senescence, ultimately leading to their death [54]. Consequently, for this study, we selected six-year-old Moso bamboo culms to investigate culm morphological variations in sixteen Moso bamboo populations.

### 3.2. Culm Morphological Variations among the Populations

Considering the variation among the populations, a systematic analysis of the culm morphological variations unveiled new insights into culm morphogenesis. The significant variability in the culm heights among the populations (Appendix A) suggests diverse genetic adaptations of the populations to various environmental conditions and their influences on the culm heights [8,47]. The measurement of the culm heights under the first branch demonstrated substantial variation, both in the height and in their relative contribution to the total culm height (Table 1, Appendix A), indicating the significance of the culm heights under the first branch. Consequently, the culm heights under the first branch were established as the standard for further analysis. This is consistent with a previous report on clumping bamboo (*Thyrsostachys oliveri*), where culm heights under the first branch were considered standard materials for investigating morphological characteristics [55]. Furthermore, our study observed the internode patterns, such as the lengths and diameters, aligning with a previous study on clumping bamboo [55], which revealed that the internode length, diameter, and wall thickness vary along the culm length, indicating distinct patterns.

Notably, the study found that in clumping bamboo [55], the internode length increased with the culm height up to the 20th internode, while the diameter and wall thickness exhibited a contrasting trend. Specifically, the thickness decrease with the culm height increase was more distinct compared to the diameter, with the largest internode volume observed in the 7th to 10th internodes. In our study, we also observed that the largest internode volume, found in the 9.5th to 14.9th internodes across all the populations (Figure 4), follows a distinct pattern. Initially, it decreases at the bottom of the culm, increases towards the middle, and subsequently decreases again towards the top as the internode order, indicating the position along the culm height, increases. Furthermore, our previous study [45] reported that the reduced total number of internodes, culm diameter, and internode length significantly decreased the total culm height in Moso bamboo. Consistently, our current study observed a similar pattern with the total culm height, suggesting that the total number of internodes, culm diameter, and internode length determine culm heights among all the populations.

The population CS exhibited the highest average culm height (Appendix A) and also stood out with the highest average internode wall thickness (Appendix A), indicating its potential for various industrial applications. Furthermore, among the populations, only the population CS consistently demonstrates superior values across multiple parameters. A previous study revealed significant variations in culm traits among three different Moso bamboo populations from Anhui, Guangxi, and Zhejiang provinces, with the Anhui population exhibiting superior values, followed by the Guangxi population [47]. The current study also encompassed five distinct populations from these three provinces: HS and JZ from Anhui, XA from Guangxi, and AJ and LY from Zhejiang. The population AJ displayed the lowest culm height under the first branch, measuring 4.91 m (Appendix A), indicating shorter overall growth. In contrast, the population JZ exhibited the highest variability in culm heights under the first branch (Appendix A), suggesting a wider range of height distribution within the population. Additionally, the populations JZ and LY showed the lowest ratios of culm heights under the first branch to total culm height. Furthermore, the populations HS, JZ, and LY had moderate wall thicknesses, while populations AJ and XA had thinner walls (Appendix A), suggesting a wider range of thickness among the populations.

### 3.3. Both Diameter and Thickness Significantly Contribute to Culm Development

The analysis revealed both positive and negative correlations among the 17 culm parameters analyzed. Both positive and negative correlations were identified among these parameters (Table 1), indicating tendencies for these parameters to increase or decrease together. The diameter and thickness slope had more negative correlations. Notably, while both the diameter and thickness contribute significantly to the culm height and internode volume, the analysis suggests that the diameter may have a slightly stronger association with the culm height. For instance, the internode diameter was positively correlated with the internode thickness but had a strong negative correlation with the diameter slope. The culm height had positive correlations with the diameter, thickness, height under the first branch, internode volume, and CVHUFB. However, our study observed no significant correlation between the thickness and the height under the first branch. This is inconsistent with a previous report on Moso bamboo [47], as there are varying correlations among the traits, both in phenotype and heredity. The parameters such as diameter at breast height (DBH), node length at DBH, wall thickness at DBH, clear height, average of nodes length under the first branch, and nodes under the first branch were deemed significant for culm development. In our study, the parameters examined differ from those in the previous study, which included phenotypic traits, cell structure properties, and material properties [47]. Furthermore, our study is the first to report the correlation of 17 culm parameters, with a specific focus on each internode under the first branch, among 16 different populations in China.

### 3.4. Moso Bamboo Exhibits a Structural Resemblance to Building Construction Patterns

The correlation analysis reveals that the culm parameters tend to increase or decrease together, collectively impacting the culm height. Notably, while both the internode length and internode order demonstrate a strong positive correlation (Figure 1), the internode diameter and thickness exhibit a negative correlation with the order (Figure 2 and Appendix A), suggesting a decrease as the internode order increases. This pattern resembles building construction, where height increases while weight decreases with the addition of floors. Moreover, an interesting finding is that when the internode diameter increases, the internode length decreases, while the internode thickness increases (Figure 5). This inverse relationship suggests a trade-off mechanism in bamboo growth, where variations in one parameter drive compensatory changes in others, ensuring structural integrity and balance. Similarly, the relationship between the internode volume and order highlights variations in size along the culm height, with distinct patterns observed (Figure 4). Furthermore, consistent variation in internode order suggests a uniform pattern of internode development along the culm height (Figure 4). Additionally, strong correlations exist between the internode length, thickness, and diameter, with varying rates of change denoted by hillslope values (Figure 5 and Figure 6). These observations suggest that Moso bamboo demonstrates a gradual decline and increase in parameters, akin to building construction, with a robust base supporting the structure as it grows taller. This is consistent with previous reports on Moso bamboo [8], *Bambusa multiplex* [34], and clumping bamboo [55], where the internode length, diameter, and thickness change with the culm height. Additionally, similar to our study (Figure 4), the internode volume initially decreases at the bottom, increases towards the middle, and decreases again towards the top of the culm as the internode order, indicating the position along the culm height. This underscores the importance of diameter and thickness changing with the culm height in maintaining structural integrity and strength throughout bamboo growth.

### 3.5. Monthly Precipitation Negatively Regulates Internode Diameter and Thickness

Previous studies on Moso bamboo have shown that environmental factors influence the internode length, mechanical properties, and culm development [8,47]. However, to date, no study has reported how individual internode parameters are affected by environmental factors. Therefore, in this study, we aimed to understand the influence of environmental factors on each internode parameter. Our study observed that monthly temperature and humidity exhibited no significant correlation with both the internode diameter and thickness (Figure 7a,c). Interestingly, we found that monthly precipitation in December and January negatively regulated both the internode diameter and thickness. Furthermore, we hypothesize that environmental factors may influence the primary thickening growth, as we observed a negative impact of monthly precipitation on the internode diameter and thickness. Since the primary thickening growth occurs underground from December to January, precipitation during this typical period (Figure 7b) could also affect both the internode diameter and thickness. The primary thickening growth is an essential process [32], where the bud becomes active from the dormant stage and develops into a mature shoot with distinguishable internodes and nodes, and the pith cavity formation also occurs simultaneously. This process plays an important role in determining the internode sizes and thereby affects both the internode diameter and thickness, as well as the overall culm development [31,32,45]. Consequently, further identifying key genes that control crucial traits, such as the primary thickening growth, internode diameter, and internode thickness, is essential for exploring the molecular mechanisms underlying those traits.

## 4. Materials and Methods

### 4.1. Plant Materials and Sampling Regions

In accordance with our prior study experience in sample collection [8], we collected culm samples from various Moso bamboo populations across 16 sampling regions in China. Each sampling region represents a distinct Moso bamboo population and is identified as one of the major bamboo-producing areas in China. Each population was assigned an abbreviation code, such as AJ, CS, CY, DA, HS, JZ, LY, RH, TJ, WYS, XA, XN, YA, YF, YX, and ZT, corresponding to the 16 sampling regions in China. The detailed information about the sampling regions, along with a geographical map presenting the populations’ distribution, is provided in Appendix A. The distribution of Moso bamboo in China covers the Qinling Mountains, the Hanshui River basin to the south of the Yangtze River basin, Taiwan Province, and the Yellow River basin. The complex terrain and climate in these areas impact the growth and yield of Moso bamboo, with notable variations in climate and soil properties observed across the different regions. The altitude of these areas ranges from 34 to 1100 m, with distinct variations observed between each region, covering an area of approximately 300 acres or more. Average annual precipitation varies from 843 to 1927 mm, and annual mean temperatures fluctuate between 15.0 and 19.6 °C, with a minimum of −11 °C and a maximum of 40 °C, respectively.

### 4.2. Morphological Analysis

To analyze culm morphological variations among Moso bamboo populations, six-year-old Moso bamboo culms were selected for this study. Thirty uniformly healthy culms with good growth status were randomly chosen from each sampling region. Initially, the diameter at breast height (DBH) (measured at 1.3 m from the ground) was recorded, and the average value was calculated. Based on the average DBH value, thirty additional culms were selected and then cut down, and various morphological parameters were measured. These parameters included culm height (CH), culm height under the first branch (HUB), total number of internodes, average internode number (AIN), average internode length (AIL), average internode thickness (AIT), average internode diameter (AID), average internode length under the first branch, average internode wall thickness under the first branch, and average internode diameter under the first branch. The measurements were taken from the ground to the first branch of the culm when referring to “under the first branch”. The average values for each measurement were then calculated.

### 4.3. Statistical Analysis

The statistical analysis of culm morphological measurements involved several conventional mathematical methods, including basic statistical analysis, regression analysis, and Pearson correlation analysis. Data processing and statistical analyses were performed using SPSS statistics V22.0 software (https://www.ibm.com/products/spss-statistics (accessed on 14 March 2024)) [56] and GraphPad Prism 8.0 (GraphPad Software, San Diego, CA, USA, www.graphpad.com). The analysis specifically included exploring the correlation among internode diameter, internode length, and bamboo wall thickness across different Moso bamboo populations. The data are presented as mean ± standard deviation (SD). All figures were created using GraphPad Prism 8.0, and the statistical significance was defined at *p* < 0.05.

## 5. Conclusions

In conclusion, our study observed significant differences in the culm heights and other parameters among the populations, indicating genetic diversity and distinct characteristics. Interestingly, the culm heights under the first branch showed higher variability compared to the total culm heights among the populations and within each population. The culm parameters demonstrated a tendency to increase or decrease together across all the populations, significantly influencing the culm height. Additionally, the increase in the internode wall thickness is the result of secondary wall thickening, characterized by a multi-layered structure with alternating width [8], emphasizing the importance of thickness in bamboo mechanical properties. This thickness exhibits a positive correlation with the internode diameter; thus, both the internode diameter and thickness play crucial roles in determining the culm height and its mechanical properties. However, monthly precipitation in December and January had a negative impact on both the internode diameter and thickness, suggesting that precipitation may negatively impact the primary thickening growth during these months, affecting both the internode diameter and thickness. Overall, our study emphasizes the importance of understanding the culm morphology and its underlying genetic and environmental determinants for effective bamboo cultivation and utilization. Further research on identifying key genes controlling these traits will be essential for improving bamboo productivity and resilience in diverse environments.

## Figures and Tables

**Figure 1 plants-13-01484-f001:**
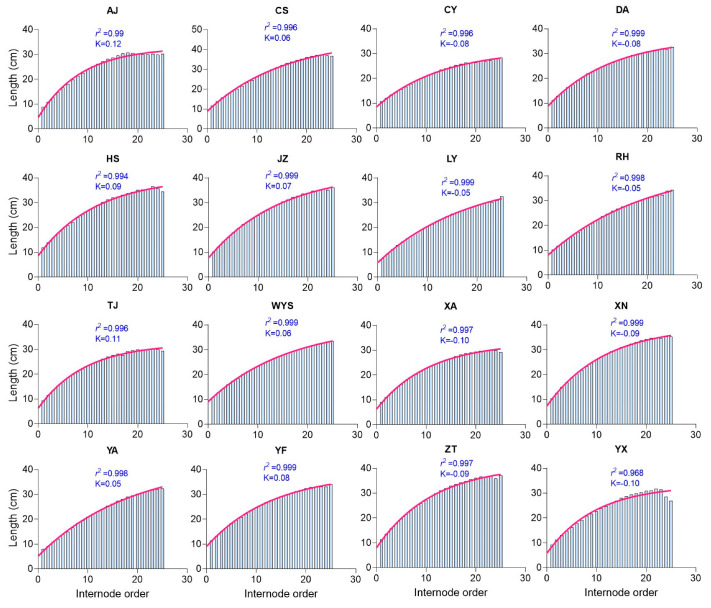
The coefficient of determination (*r*^2^) illustrates the strong positive correlation between the internode length and internode order across various Moso bamboo populations. Each abbreviation code, such as AJ, CS, CY, etc., represents 16 sampling regions in China. The *r*^2^ value represents the coefficient of determination, indicating the strength of the correlation, while the K value represents the rate constant expressed in inverse units of the variable represented on the *X*-axis.

**Figure 2 plants-13-01484-f002:**
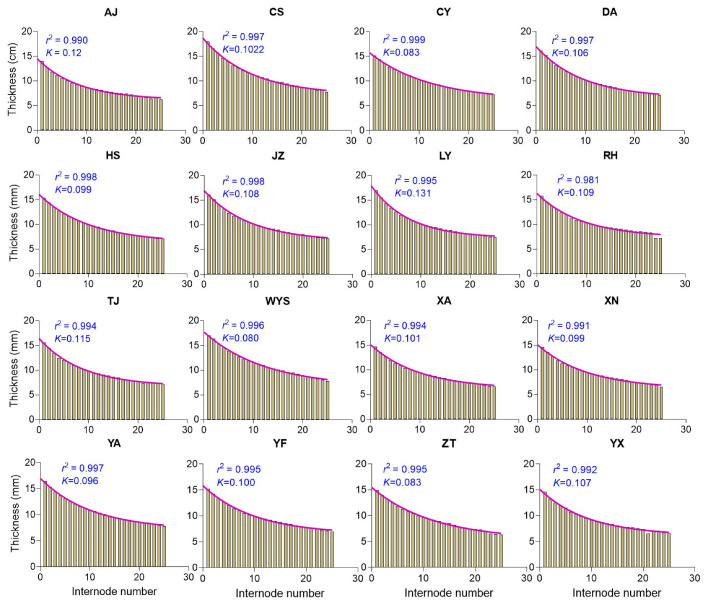
The correlation analysis illustrates the relationship strength (*r*^2^) between the internode thickness and internode order across various Moso bamboo populations. Each abbreviation code, such as AJ, CS, CY, etc., represents 16 sampling regions in China. The *r*^2^ value represents the coefficient of determination, indicating the strength of the correlation, while the K value represents the rate constant expressed in inverse units of the variable represented on the *X*-axis.

**Figure 3 plants-13-01484-f003:**
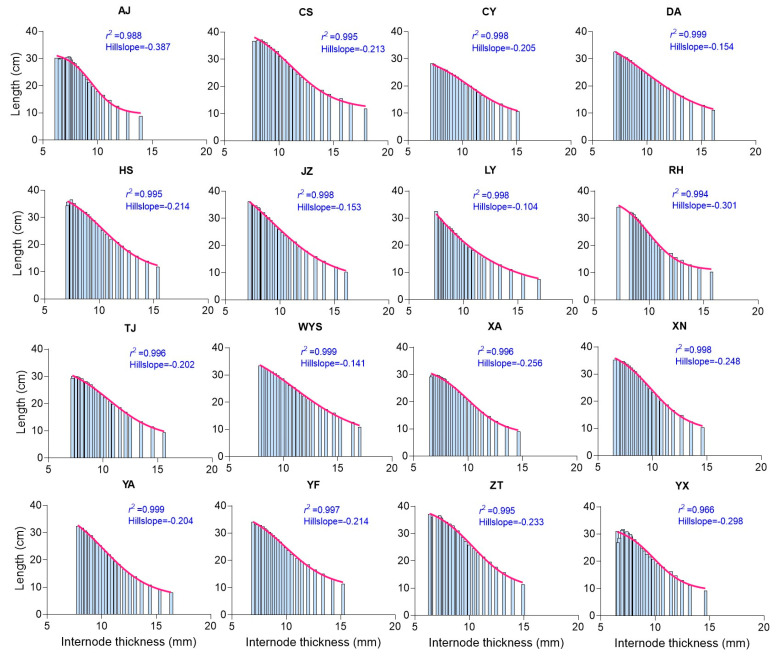
The correlation analysis illustrates the relationship strength (*r*^2^) and hillslope values between the internode thickness and internode length across various Moso bamboo populations. Each abbreviation code, such as AJ, CS, CY, etc., represents 16 sampling regions in China. The *r*^2^ value represents the coefficient of determination, indicating the strength of the correlation, while the negative hillslope value indicates the rate of change in the dependent variable (internode thickness) for a one-unit increase in the independent variable (internode length). For example, the hillslope value of −0.387 indicates that for every one-unit increase in internode thickness, the internode length decreases by 0.387 units on average, as observed in the AJ population.

**Figure 4 plants-13-01484-f004:**
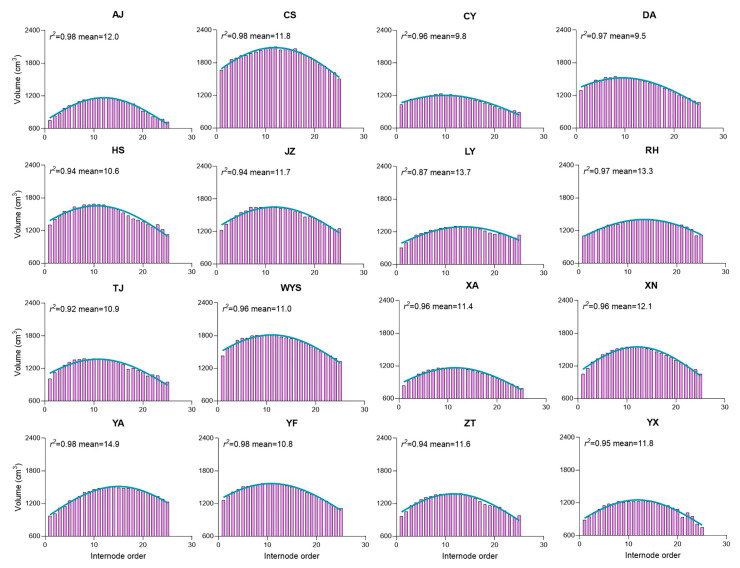
The coefficient of determination (*r*^2^) illustrates the relationship between the internode volume and internode order, suggesting changes in the internode volumes along the culm height across various Moso bamboo populations. Each abbreviation code, such as AJ, CS, CY, etc., represents 16 sampling regions in China. The *r*^2^ value indicates the strength of the correlation, while the mean internode order value illustrates that the highest average internode volume occurs at the middle of the culm. For example, in the AJ population, the highest average internode volume occurs at the 12th internode.

**Figure 5 plants-13-01484-f005:**
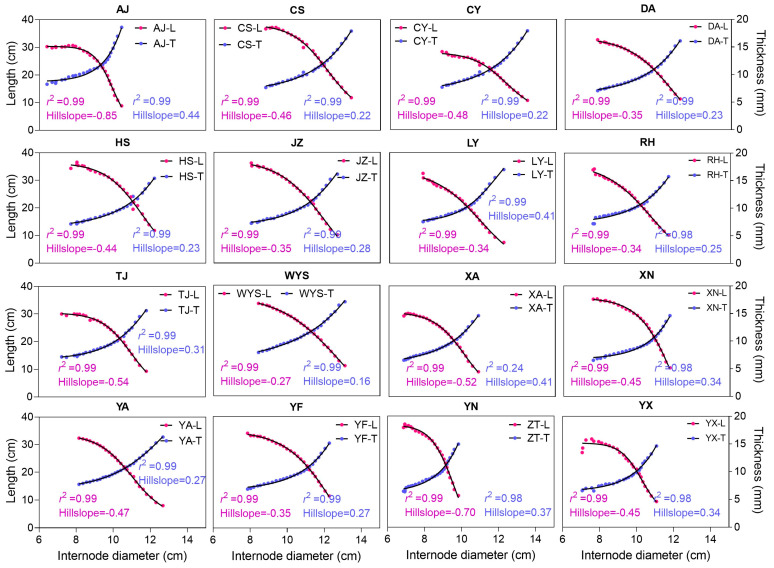
The correlation coefficients and hillslope values illustrate the relationship between the various parameters, such as the internode length, thickness, and diameter, across various Moso bamboo populations. Each abbreviation code, such as AJ, CS, CY, etc., represents 16 sampling regions in China. The *r*^2^ value indicates the correlation coefficient, reflecting the strength of the correlation, while the hillslope value indicates the rate of change between variables. A positive hillslope value signifies an increasing trend, whereas a negative value indicates a decreasing trend, as observed in the AJ population. The letters L and T on the right side of each graph represent internode length (cm) and thickness (mm), respectively. The graph line in purple-blue represents the relationship between the internode diameter and thickness, while the red line represents the relationship between the internode diameter and length, with the internode diameter represented on the *X*-axis.

**Figure 6 plants-13-01484-f006:**
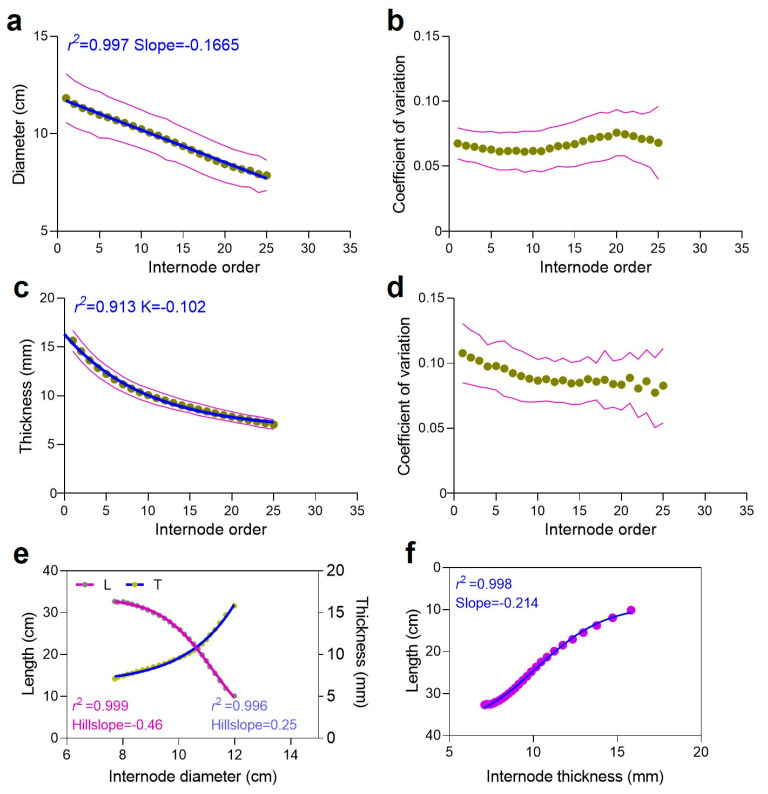
The coefficient of determination (*r*^2^) (regression analysis) among different internode parameters in Moso bamboo, collected from 16 Moso bamboo populations in China. The internode parameters demonstrate the dynamic patterns of internode morphology, thereby influencing culm height. (**a**) The relationship between the internode order and internode diameter. (**b**) The coefficient of variation of the internode order with respect to (**a**). (**c**) The relationship between the internode order and internode thickness. (**d**) The coefficient of variation of the internode order with respect to (**c**). (**e**) The relationship between the internode diameter and internode length, and the relationship between the internode diameter and internode thickness. The letters L and T on the top represent internode length (cm) and thickness (mm), respectively. The graph line in purple red represents the relationship between the internode diameter and length, while the purple blue line represents the relationship between the internode diameter and thickness, with the internode diameter represented on the *X*-axis. (**f**) The relationship between the internode thickness and internode length. The *Y*-axis represents, from top to bottom, 0–40 (cm). The *r*^2^ value represents the coefficient of determination, indicating the strength of the correlation, while the slope and hillslope values indicate the rate of change in the dependent variable (e.g., internode diameter) for a one-unit increase in the independent variable (e.g., internode order). For example, the slope value of −0.1665 indicates that for every one-unit increase in the internode order, the internode diameter decreases by 0.1665 units on average, as observed in (**a**).

**Figure 7 plants-13-01484-f007:**
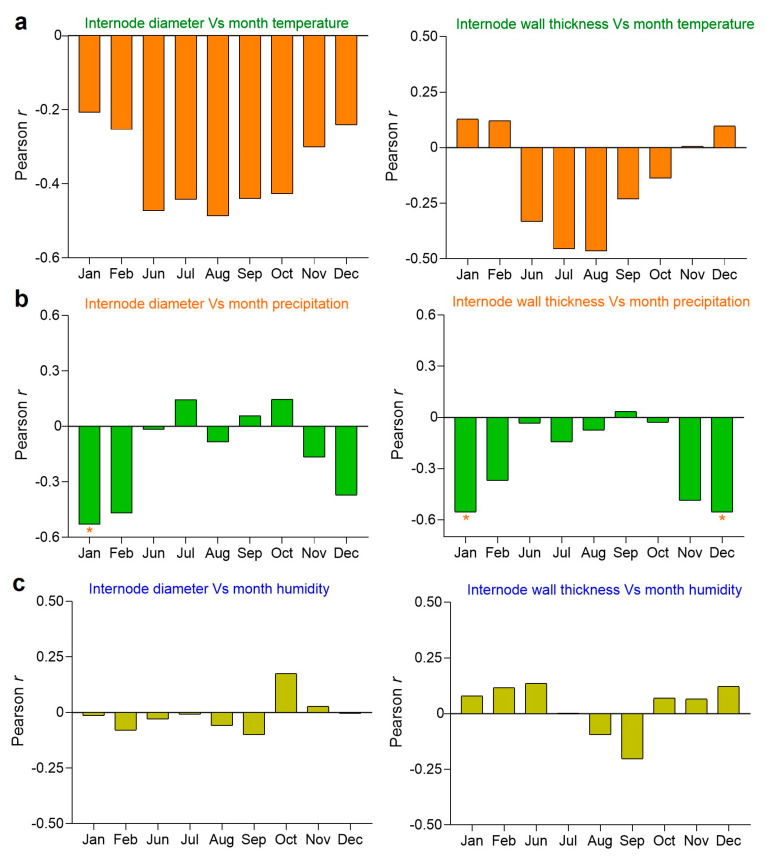
The Pearson correlation between environmental factors and the internode diameter and thickness of Moso bamboo, collected from 16 Moso bamboo populations in China. (**a**) The internode diameter with monthly temperature, and the internode thickness with monthly temperature. (**b**) The internode diameter with monthly precipitation, and the internode thickness with monthly precipitation. (**c**) The internode diameter with monthly humidity, and the internode thickness with monthly humidity. The positive values (such as 0.3 or 0.6) and negative values (such as −0.3 or −0.6) in the Pearson r value represent positive and negative correlations, respectively. The symbol (*) represents significant correlations defined at *p* < 0.05.

**Table 1 plants-13-01484-t001:** The correlation analysis between different culm parameters in Moso Bamboo. Each column represents a distinct parameter collected from 16 Moso bamboo populations in China.

	*D*	*T*	*D Slope*	*T Slope*	*DT Slope*	*DL Slope*	*TL Slope*	*L*	*H*	*HUFB*	*INUFB*	*IV*	*CVHUFB*	*Ratio FBRS*	*Vmax* *IS*	*Vmax in ratio*	*Vmax H ratio*
*D*	1																
*T*	0.69 *	1															
*D Slope*	−0.85 *	−0.37	1														
*T Slope*	0.04	−0.23	0.47	1													
*DT Slope*	−0.62 *	−0.67 *	−0.61 *	0.54 *	1												
*DL Slope*	0.62 *	0.54 *	−0.38	−0.07	−0.62 *	1											
*TL Slope*	0.52 *	0.64 *	−0.23	−0.11	−0.42	0.67 *	1										
*L*	0.48	0.27	−0.43	−0.38	−0.39	0.10	0.07	1									
*H*	0.71 *	0.66 *	−0.45	−0.41	−0.74	0.65 *	0.50 *	0.66 *	1								
*HUFB*	0.63 *	0.47	−0.39	−0.35	−0.66 *	0.42	0.25	0.69 *	0.87 *	1							
*INUFB*	0.44	0.40	−0.60	−0.25	−0.56 *	0.52 *	0.35	0.18	0.68 *	0.81 *	1						
*IV*	0.89 *	0.75 *	−0.62 *	−0.23	−0.66 *	0.46	0.43	0.77 *	0.82 *	0.74 *	0.39	1					
*CVHUFB*	0.86 *	0.75 *	−0.04	−0.26	−0.75 *	0.54 *	0.43	0.64 *	0.89 *	0.90 *	0.70 *	0.92 *	1				
*Ratio FBRS*	−0.12	−0.19	0.06	0.14	0.004	−0.27	−0.57 *	0.14	−0.11	0.32	0.28	−0.05	0.11	1			
*Vmax IS*	−0.04	−0.01	0.40	0.34	0.33	−0.02	−0.11	−0.08	0.07	0.23	0.41	−0.06	0.11	0.38	1		
*Vmax in ratio*	−0.41	−0.34	0.44	0.54 *	0.81 *	−0.48	−0.41	−0.23	−0.52 *	−0.47	−0.45	−0.39	−0.49	0.15	0.63 *	1	
*Vmax H ratio*	−0.41	−0.33	0.47	0.48	0.80 *	−0.49	−0.30	−0.17	−0.53 *	−0.53 *	−0.54 *	−0.36	−0.52 *	0.001	0.50 *	0.96 *	1

* Significant at the *p* < 0.05. D (internode diameter); T (internode thickness); D Slope (diameter slope); T Slope (thickness slope); DT Slope (diameter and thickness slope); DL Slope (diameter and length slope); TL Slope (thickness and length slope); L (internode length); H (culm height); HUFB (height under the first branch); INUFB (internode number under the first branch); IV (internode volume); CVHUFB (coefficient of variation of height under the first branch); Ratio FBRS (ratio of first branch to culm height); Vmax IS (maximum volume internode number); Vmax in ratio (maximum volume internode number in ratio); and Vmax H ratio (maximum volume height between nodes).

## Data Availability

The data that support the findings of this study are available from the corresponding authors upon reasonable request.

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
