# Peer review of "Culm Morphological Analysis in Moso Bamboo Reveals the Negative Regulation of Internode Diameter and Thickness by Monthly Precipitation"

_plants, 2024, doi:10.3390/plants13111484_

Round 1
Reviewer 1 Report
Comments and Suggestions for Authors
The present reviewer does not know bamboo species and their patterns of variation well and has a major scientific competence in ecological, quantitative and population genetics of forest trees. This is the basis for the present evaluation.
General comments
The manuscript presents a detailed study of the morphological variation in culm traits among 16 sampling regions, called populations, of one major bamboo species in China, Moso. Many culm morphological traits have been measured, and population means, ranges of variation within populations and statistical measures have been calculated and analysed. Correlation coefficients between traits have been calculated and between traits means and environmental factors. All these numbers are presented in tables and in figures. All tables and figures are discussed in detail, with repeated presentations of population means of traits.
It is the opinion of this reviewer that too many details are presented and with repetitions. This is not following the journal instructions, saying: “Provide a concise and precise description of the experimental results, their interpretation as well as the experimental conclusions that can be drawn.” The manuscript in its present form should therefore not be published in an internation journal. It could be published as a more technical report, which may be of importance when bamboo materials with specific structural patterns are sought.
I therefore do not recommend that the manuscript in its present form should be published in this journal. A more precise and shorter version could be prepared, more focused on the relationships indicated in the title.
Specific comments
L. 61 – 71: Which in-depth study, write the reference earlier in the paragraph?
L. 72 – 77: Define the study initially in this paragraph.
L. 132 – 134: The CS population had not the highest culm height under the first branch, see Table 2.
L. 215-217: There is no information amount genetic diversity in this article. It only precents phenotypic variation among individuals and among populations.
L. 230-232: Same comment as above.
L. 769-829: Only monthly temperatures, precipitation and humidity are studied. The populations have altitudinal range from 34 to 1100 m. Are there differences in soil fertility and site index that could influence the culm morphological traits. Any relationship to altitude?
P. 986 – 1001: The populations are not clearly defined. It is defined as a sampling region. How large is each area? What was the distance between the trees? Could each sampling area be defined as a population in the genetic sense? A map presenting the range of the species in China and a more the distribution of the populations would be useful for readers outside China.
P. 1017 – 1027: Define the statistical model used for the analyses of variance. The sentence on lines 1018 – 1019 is not precise. Has factor analysis been used, as stated?
Author Response
Dear Reviewer,
Thank you very much for dedicating your time to review our manuscript. Enclosed, please find our detailed responses to your comments in the document titled "Author response to Reviewer_1_plants-2941524" and the corresponding revisions highlighted in the file "Plants-2941524_R1 with track changes.pdf".
Sincerely,
Professor Qiang Wei
Dr. Muthusamy Ramakrishnan

Reviewer 2 Report
Comments and Suggestions for Authors
Dear Authors,
below are my comments:
The abstract is too long - please shorten it to a maximum of 200 words, as requested by the MDPI.
The References are not formatted correctly. This should be corrected. The Literature Introduction is adequate, but there are too few references, barely more than 20. I therefore ask the authors to significantly expand the chapter and please use mainly results published in the last 5 years. The authors should take this seriously.
Results. I suggest other illustrative tools because these rotated, large number of tables degrade the quality of the manuscript. Please use others, or use these in other formats. This is not appropriate.
Diagrams: include only the necessary ones, the rest should be in annexes.
Results. I suggest other illustrative tools because these rotated, large number of tables degrade the quality of the manuscript. Please use others, or use these in other formats. This is not appropriate.
Author Response
Dear Reviewer,
Thank you very much for dedicating your time to review our manuscript. Enclosed, please find our detailed responses to your comments in the document titled "Author response to Reviewer_2_plants-2941524" and the corresponding revisions highlighted in the file "Plants-2941524_R1 with track changes.pdf".
Sincerely,
Professor Qiang Wei
Dr. Muthusamy Ramakrishnan

Round 2
Reviewer 1 Report
Comments and Suggestions for Authors
Figure 7 is misplaced in Section 3.1, should be in Section 2.10 where the results are discussed. The figure legend is written both in lines 612 - 618 and lines 674-680.
Author Response
Comment 1:
Figure 7 is misplaced in Section 3.1, should be in Section 2.10 where the results are discussed. The figure legend is written both in lines 612 - 618 and lines 674-680.
Response:
Dear Reviewer,
We sincerely appreciate and are grateful to you for identifying the misplaced figure. We agree that Figure 7 is more appropriate in Section 2.10 "Results" for better context. We have relocated the figure to the Section 2.10. We hope that the revised manuscript has been significantly improved and is now suitable for publication in Plants. We sincerely appreciate your valuable input, which has helped us improve the manuscript.
Comment 2:
The figure legend is written both in lines 612 - 618 and lines 674-680.
Response:
We thank the reviewer for pointing out the potential issue with a duplicate figure legend. We apologize for any confusion, but we couldn't locate the legends in the mentioned lines (612-618 and 674-680) of our manuscript. To ensure clarity, we have carefully checked the manuscript for any duplicate legends associated with Figure 7 (or any other figures). We would be happy to make any necessary corrections if the reviewer could kindly provide additional information on the specific content found in those lines.
Sincerely yours
Professor Qiang Wei
Dr. Muthusamy Ramakrishnan
Reviewer 2 Report
Comments and Suggestions for Authors
Dear Authors,
I have reviewed the manuscript and make the following comments: I accept the manuscript for publication.
Author Response
Comment 1:
I have reviewed the manuscript and make the following comments: I accept the manuscript for publication.
Response:
Dear Reviewer,
We sincerely thank you for your positive feedback and endorsement of our manuscript for publication in Plants. We greatly appreciate the time and effort you invested in reviewing our work. Your acceptance is highly encouraging and motivates us to continue our research in this field.
Sincerely yours
Professor Qiang Wei
Dr. Muthusamy Ramakrishnan